# Loss of bone morphogenetic protein-binding endothelial regulator causes insulin resistance

Hua Mao[1,2], Luge Li[1,2], Qiying Fan[1,2], Aude Angelini [1,2], Pradip K. Saha[3], Huaizhu Wu[1,2], Christie M. Ballantyne [1,2], Sean M. Hartig [3,4], Liang Xie[1,2] & Xinchun Pi [1,2✉]

Accumulating evidence suggests that chronic inflammation of metabolic tissues plays a causal role in obesity-induced insulin resistance. Yet, how specific endothelial factors impact metabolic tissues remains undefined. Bone morphogenetic protein (BMP)–binding endothelial regulator (BMPER) adapts endothelial cells to inflammatory stress in diverse organ micro-environments. Here, we demonstrate that BMPER is a driver of insulin sensitivity. Both global and endothelial cell-specific inducible knockout of BMPER cause hyperinsulinemia, glucose intolerance and insulin resistance without increasing inflammation in metabolic tissues in mice. BMPER can directly activate insulin signaling, which requires its internalization and interaction with Niemann-Pick C1 (NPC1), an integral membrane protein that transports intracellular cholesterol. These results suggest that the endocrine function of the vascular endothelium maintains glucose homeostasis. Of potential translational significance, the delivery of BMPER recombinant protein or its overexpression alleviates insulin resistance and hyperglycemia in high-fat diet-fed mice and Lepr[db/db] (*db/db*) diabetic mice. We conclude that BMPER exhibits therapeutic potential for the treatment of diabetes.

[1] Department of Medicine, Section of Athero & Lipo, Baylor College of Medicine, Houston, TX, USA. [2] Cardiovascular Research Institute, Baylor College of Medicine, Houston, TX, USA. [3] Department of Medicine, Division of Diabetes, Endocrinology & Metabolism, Diabetes Research Center, Baylor College of Medicine, Houston, TX, USA. [4] Departments of Molecular and Cellular Biology, Baylor College of Medicine, Houston, TX, USA. ✉email: xpi@bcm.edu

Diabetes mellitus (DM) is a group of metabolic diseases characterized by chronic hyperglycemia and impaired carbohydrates, lipids, and protein metabolism resulting from defects in insulin secretion, insulin action, or both. More than 100 million U.S. adults are now living with diabetes or prediabetes, a condition that if not treated often leads to type 2 diabetes mellitus (T2DM) within 5 years[1]. T2DM is the most common form of DM, accounting for 90 to 95% of all diabetic patients, and is expected to increase to 693 million by 2045[2]. T2DM is associated with an increased risk of micro- and macrovascular complications, such as diabetic nephropathy, neuropathy, retinopathy, and coronary artery disease, which impose profound impacts on the quality of life and health care resources. Current management of diabetes includes lifestyle intervention, routine blood glucose monitoring, and pharmacological therapy. However, high costs, variable efficiency, and tolerability are important barriers to the use of these pharmacological treatments.

Insulin resistance is a core defect of T2DM and is associated with chronic inflammatory responses. During chronic inflammation, macrophage plays a crucial role in obese-induced insulin resistance[3,4]. After microphages are infiltrated into metabolic tissues such as obese adipose tissue, their secreted proinflammatory cytokines inhibit insulin signaling and result in insulin resistance. Another key player of chronic inflammation is vascular endothelium, which coordinates the action of immune cells through the continual adjustment in its own structure and function, including induction of adhesion molecules and proinflammatory cytokines, disruption of its barrier function, and angiogenesis[5]. Even though endothelial dysfunction and inflammation have been observed during the development of T2DM[6,7], it remains unknown whether endothelial cell (EC) plays a crucial role in the maintenance of glucose homeostasis and how its dysregulation contributes to insulin resistance and diabetes.

BMP pathway is known as an important regulator in the maintenance of endothelial integrity and the induction of vascular inflammatory responses[8]. BMPER (also called CV-2) binds BMPs and is an extracellular modulator of BMP signaling pathway[9]. We have discovered BMPER plays a pivotal role in BMP-mediated endothelial events and chronic inflammation in the vasculature[10–15]. However, the role of BMPER in obesity and insulin resistance has not been studied before.

In this study, we have made a surprising observation that BMPER regulates glucose homeostasis without affecting chronic inflammation. Instead, BMPER is a driver of insulin sensitivity through activating insulin signaling pathway. In hepatocytes, this activation requires BMPER endocytosis and interaction with Niemann-Pick C1 (NPC1), an integral membrane protein that transports intracellular cholesterol[16,17]. Both whole-body and EC-specific BMPER inducible knockout (iKO) mice display similar glucose defects, including hyperinsulinemia, insulin resistance, and glucose intolerance, suggesting vascular endothelium plays a causal role in glucose homeostasis through secreting metabolic regulators. More importantly, the delivery of BMPER recombinant protein or its adeno-associated viral (AAV) particles dramatically alleviates insulin resistance and hyperglycemia in diabetic mice. Our data reveal that BMPER is a protective regulator of glucose homeostasis and could become a potential therapeutic target for treating diabetes.

## Results

**Loss of BMPER results in glucose dysregulation.** BMPER null mice die at birth due to lung defects[12], which limits our understanding of BMPER's functions during pathological conditions at adulthood. To study the role of BMPER in glucose homeostasis,

**Table 1 Metabolic parameters.**

| Parameters | WT | iKO |
|---|---|---|
| BW (g) | 32.25 ± 0.86 | 32.18 ± 1.05 (NS) |
| **Serum parameters** | | |
| Glucose (mg/dl) | | |
| Fasted | 139.0 ± 4.79 | 150.0 ± 3.81 (NS) |
| Fed | 174.7 ± 6.12 | 176.4 ± 8.134 (NS) |
| Insulin (ng/ml) | | |
| Fasted | 1.31 ± 0.13 | 2.80 ± 0.22 $P=0.0002$ |
| Fed | 2.81 ± 0.20 | 16.77 ± 0.42 $P<0.0002$ |
| TG (mg/dl) | 73.12 ± 4.25 | 85.78 ± 3.30 $P=0.03$ |
| FFA (mEq/l) | 0.78 ± 0.10 | 0.78 ± 0.05 (NS) |
| **Liver parameters** | | |
| TG (mg/g tissue) | 9.81 ± 0.65 | 12.44 ± 0.69 $P=0.02$ |
| DAG (nmol/mg tissue) | 2.14 ± 0.50 | 4.53 ± 0.47 $P=0.006$ |
| BCAA (nmol/mg tissue) | 0.44 ± 0.08 | 0.58 ± 0.05 (NS) |
| Ceramide (nmol/mg tissue) | 0.15 ± 0.01 | 0.16 ± 0.001 (NS) |
| **Skeletal muscle parameters** | | |
| TG (mg/g tissue) | 5.19 ± 1.31 | 8.99 ± 1.50 (NS) |
| DAG (nmol/mg tissue) | 0.56 ± 0.08 | 0.62 ± 0.09 (NS) |
| BCAA (nmol/mg tissue) | 0.23 ± 0.02 | 0.34 ± 0.06 (NS) |
| Ceramide (nmol/mg tissue) | 0.08 ± 0.01 | 0.06 ± 0.01 (NS) |

$n = 25$ mice (**BW**, WT), 27 mice (**BW**, iKO), 6 mice (**fasted glucose and insulin**, WT), 7 mice (**fasted glucose and insulin**, iKO), 9 mice (**fed glucose**), 5 mice (**fed insulin**, WT), 6 mice (**fed insulin**, iKO), 10 mice (**TG, FFA in serum**), 7 mice (**liver and skeletal muscle parameters**, WT), 6 mice (**liver parameters**, iKO), and 5 mice (**skeletal muscle parameters**, iKO). Data are presented as mean values ± SEM. NS not significant. Analysis was unpaired two-tailed Student's $t$ test. WT, BMPER$^{flox/flox}$;CAG-CreER$^{-/-}$. iKO, BMPER$^{flox/flox}$;CAG-CreER$^{+/-}$.

we generated BMPER$^{flox/flox}$ mice using CRISPR-cas9 gene editing (Fig. 1a). A BMPER inducible knockout (iKO) mouse model was created by crossing BMPER$^{flox/flox}$ and CAG-CreER$^{+/-}$ transgenic mice (Fig. 1a), which allows temporal depletion of BMPER upon tamoxifen injection. BMPER depletion was confirmed in plasma and a variety of tissues of BMPER iKO compared to their littermate control (WT, BMPER$^{flox/flox}$; CAG-CreER$^{-/-}$) mice (Fig. 1b). Interestingly, BMPER iKO mice displayed hyperinsulinemia and higher homeostasis model assessment for insulin resistance (HOMA-IR) scores than WT mice at four months after tamoxifen injection (Fig. 1c–e, Table 1). In addition, BMPER iKO mice were more glucose intolerant and insulin resistant (Fig. 1f, g). To further explore insulin sensitivity, hyperinsulinemic-euglycemic clamp studies were performed. As shown in Fig. 1h, i, the amounts of exogenous glucose required for maintaining euglycemia (glucose infusion rate, GIR) and the glucose disposal rate (GDR) were markedly lower in BMPER iKO mice than WT mice. Both the ability of insulin to suppress hepatic glucose production, which reflects hepatic insulin sensitivity, and the insulin-stimulated glucose uptake in gastrocnemius muscle (GM), brown adipose tissue (BAT), and heart, which reflects insulin sensitivity in peripheral tissues, were significantly decreased in BMPER iKO mice (Fig. 1j, k). Next, we determined insulin-stimulated signaling in metabolic tissues and observed the activity of insulin signaling was blunted in these tissues of BMPER iKO mice, indicated by decreases in the phosphorylation of insulin receptor substrate1 (IRS1) and AKT (Fig. 1l, Supplementary Fig. 1). We also evaluated the expression of liver genes that reflect glucose output. BMPER depletion led to a significant induction of gluconeogenic enzymes, including G6Pase (glucose-6-phosphatase) and PEPCK (phosphoenolpyruvate carboxykinase), and a lipogenic transcription factor SREBP1 (sterol regulatory element-binding transcription factor 1; Supplementary Fig. 2a). However, no change was observed with the glycolytic enzyme glucokinase (GK; Supplementary Fig. 2a). Last, we

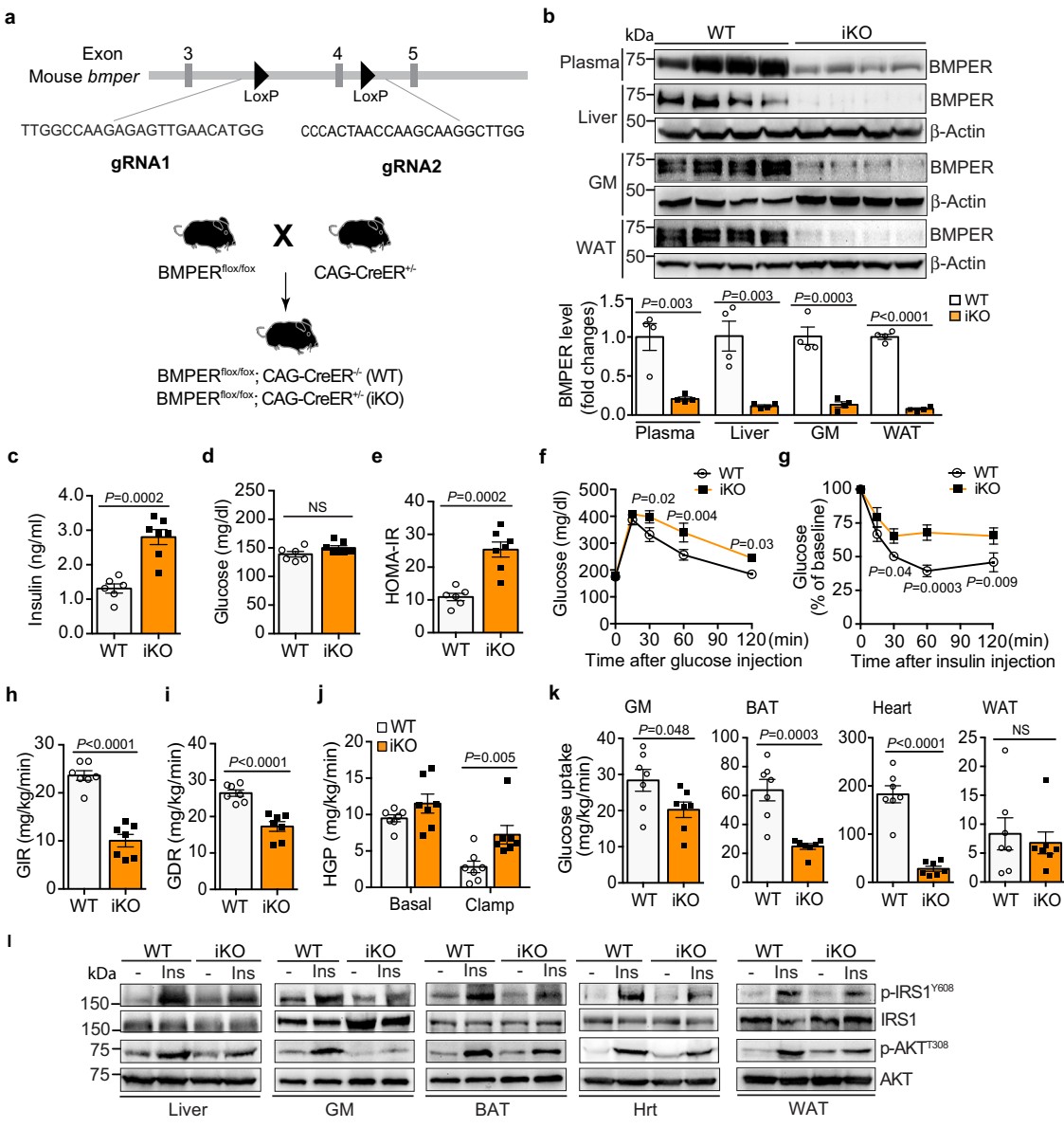

**Fig. 1 BMPER depletion causes mice to develop hyperinsulinemia, glucose intolerance, and insulin resistance. a** The generation of the BMPER iKO mouse model. The gRNA-guided CRISPR/Cas9 strategy was used for targeted deletion of *bmper* gene. **b** BMPER depletion was examined with Western blotting. **c–e** Fasted insulin and glucose, HOMA-IR. **f, g** Glucose and insulin tolerance tests. **h** Glucose infusion rate (GIR). **i** Glucose disposal rate (GDR). **j, k** Hepatic glucose production (HGP; **j**) and glucose uptake in peripheral tissues (**k**) were analyzed with hyperinsulinemic-euglycemic clamps. **l** Insulin signaling was blunted in BMPER iKO mice. Insulin (Ins, 0.5 h) was injected (*i.p.*) into BMPER iKO and WT mice. Indicated tissues were used for Western blotting. GM gastrocnemius muscle, BAT brown adipose tissue, WAT white adipose tissue, Hrt heart. WT, BMPER$^{flox/flox}$; CAG-CreER$^{-/-}$. iKO, BMPER$^{flox/flox}$; CAG-CreER$^{+/-}$. $n = 4$ mice (**b**), 6 mice (**c–e** WT), 7 mice (**c–e** iKO), 7 mice (**f, h–k**) and 5 mice (**g**). Data are presented as mean values ± SEM. NS not significant. Analysis was two-way ANOVA followed by Fisher's LSD multiple comparison test (for **f**, **g**, **j**) or unpaired two-tailed Student's *t* test (for **b–e**, **h**, **i**, **k**).

assessed contents of triglyceride (TG) and metabolites that are associated with insulin resistance. There were significant increases in plasma and liver TG contents of BMPER iKO mice (Table 1). In addition, an increase was observed with diacylglyceride (DAG) contents in BMPER iKO liver, but not with branched-chain amino acids (BCAAs) and ceramides (Table 1). Taken together, these results suggest BMPER depletion results in glucose dysregulation through regulating insulin sensitivity, gluconeogenesis, lipogenesis, and TG/DAG metabolism.

Given that vascular inflammation also disrupts the function of metabolic tissues such as liver and white adipose tissue (WAT)[18–20], we examined whether BMPER depletion resulted in an inflammatory response in these tissues. In the liver of BMPER iKO and WT mice, the induction of the inflammatory cytokine IL1β (Supplementary Fig. 2b) was not changed while others—IL6 and TNFα—were not detectable (data not shown). Their levels did not change in WAT either (Supplementary Fig. 2c). These results suggest that inflammation does not contribute to the disruption of glucose homeostasis by BMPER depletion.

BMPs play a role in obesity through regulating adipogenesis and energy storage partitioning[21–24], we asked whether BMPER also regulates body weight through modulating BMP signaling. Surprisingly, there was no body weight difference between BMPER iKO and WT mice (Table 1). We also observed no significant changes with the expression of BMPs and BMP receptor 2 (BMPR2) in the liver of BMPER iKO mice compared to WT mice (Supplementary Fig. 2d). Further CLAMS

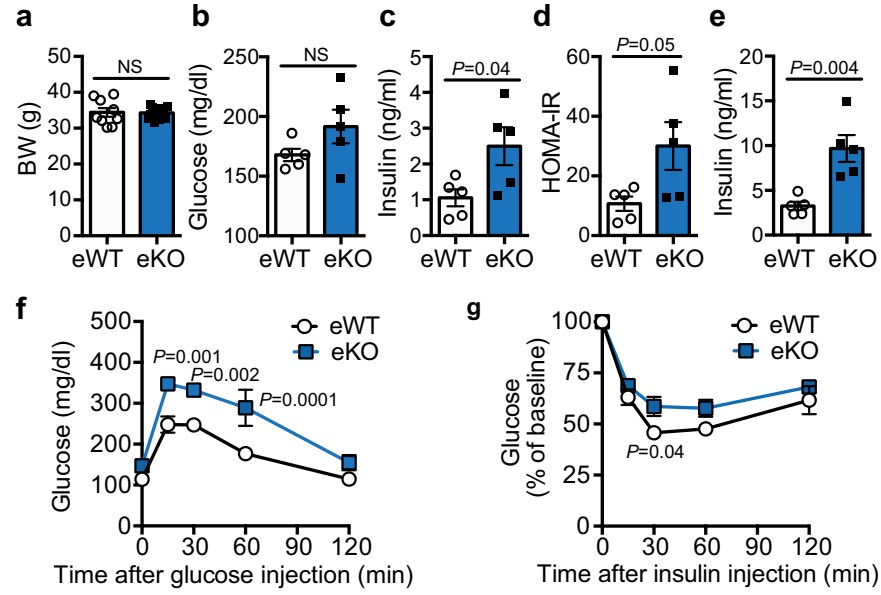

**Fig. 2 BMPER depletion in ECs leads to glucose dysregulation.** Metabolic studies were performed with BMPER eKO and eWT mice at 4 months after tamoxifen injection. **a** Body weight (BW). **b–d** Fasted glucose and insulin, HOMA-IR. **e** Fed insulin. **f**, **g** Glucose and insulin tolerance tests. eWT, BMPER^flox/flox; Cdh5-CreER^-/-. eKO, BMPER^flox/flox; Cdh5-Cre ER^+/-. n = 9 mice (**a** eWT), 10 mice (**a** eKO), 5 mice (**b–g**). Data are presented as mean values ± SEM. NS not significant. Analysis was unpaired two-tailed Student's t test (for **a–e**) or two-way ANOVA followed by Fisher's LSD multiple comparison test (for **f**, **g**).

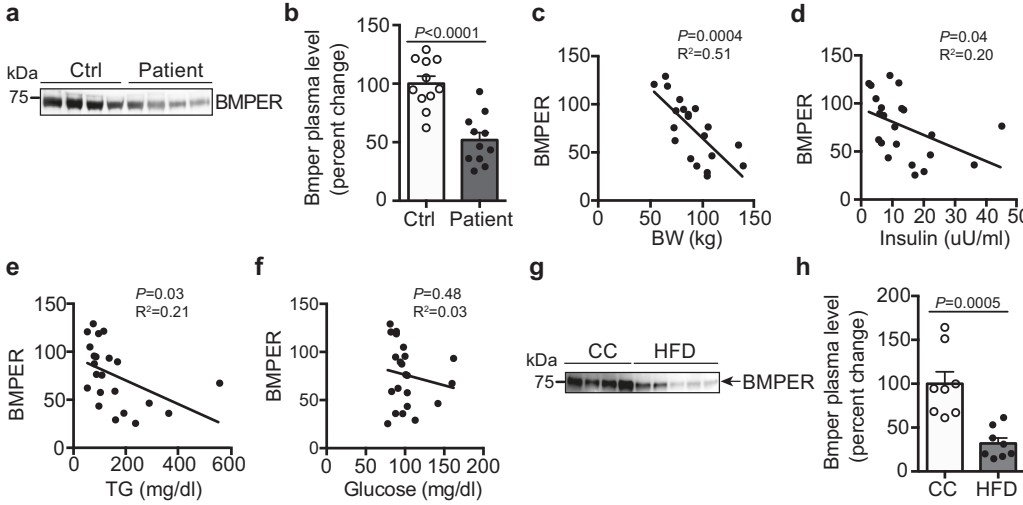

**Fig. 3 BMPER plasma level decreases in metabolic syndrome patients and DIO mice. a**, **b** BMPER plasma levels were decreased in metabolic syndrome (MS) patients[25]. **c–f** BMPER plasma level was associated with body weight (BW), insulin, and triglyceride (TG) plasma levels in MS patients. Correlation was computed for Pearson correlation coefficients with an assumption of Gaussian distribution. **g**, **h** BMPER plasma level was decreased in HFD-fed mice. n = 11 individuals (**b–f**) and 8 mice (**h**). Data are presented as mean values ± SEM. Analysis was unpaired two-tailed Student's t test (for **b**, **h**).

(comprehensive lab animal monitoring system) studies demonstrated a mild decrease in respiration exchange ratio (RER), which could be due to an increase in physical activity (Supplementary Fig. 3). No differences were detected with food and water intake of BMPER iKO mice (Supplementary Fig. 3). It suggests, unlike BMPs, BMPER might not play a significant role in body weight control.

Given that BMPER iKO mice spontaneously developed hyperinsulinemia, insulin resistance, and glucose intolerance without weight gain (Fig. 1c–l, Table 1), we hypothesized BMPER iKO mice are more sensitive to the metabolic effects of high-fat diet (HFD). Following 8 weeks of HFD feeding, BMPER iKO and WT mice gained similar weight and displayed comparable glucose

and insulin levels (Supplementary Fig. 4a–c). However, BMPER iKO mice exhibited more severe glucose intolerance and insulin resistance than WT mice (Supplementary Fig. 4d, e). These observations suggest BMPER depletion exacerbates obesity-induced insulin resistance and glucose intolerance.

**Loss of BMPER in ECs results in glucose dysregulation.** Our previous studies[10–15] suggest that BMPER is highly expressed in vascular endothelial cells (ECs). We further evaluated BMPER expression in different cells and tissues and observed that BMPER expression was highly enriched in metabolic tissues (BAT, GM, liver, and WAT; Supplementary Fig. 5a). In addition, BMPER

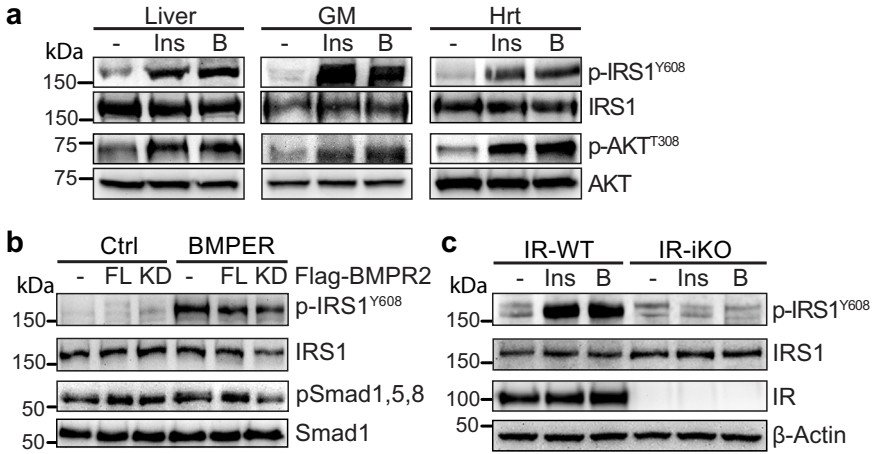

**Fig. 4 BMPER promotes insulin signaling through IR but not BMPR2. a** BMPER (B, 1 h) or insulin (Ins, 0.5 h) was injected (*i.p.*) into mice. Tissues, including liver, GM (gastrocnemius muscle), and Hrt (heart), were used for Western blotting. **b** Hepatocytes were transduced with adenovirus of flag-tagged BMPR2 full-length (FL) or kinase-dead mutant (KD). Cells were then treated with BMPER (1 h) and harvested for Western blotting. **c** Hepatocytes were isolated from IR-iKO and their littermate control (IR-WT) mice. Cells were treated with insulin (Ins, 0.5 h) or BMPER (B, 1 h) and harvested for Western blotting. IR-WT, IR[flox/flox]; CAG-CreER[−/−]. IR-iKO, IR[flox/flox]; CAG-CreER[+/−].

content was >20-fold higher at mRNA level and fourfold higher at protein level in primary ECs than hepatocytes and mouse embryonic fibroblasts (MEFs, Supplementary Fig. 5b, c). Next, we tested how metabolic stress impacts BMPER expression and detected a reduction of BMPER protein levels in EC lysates and conditioned media following treatments of high glucose, advanced glycation end products (AGEs), or palmitic acids-treated ECs (Supplementary Fig. 5d). In addition, BMPER mRNA levels were decreased by treatments of palmitic acid and high glucose, but not AGEs (Supplementary Fig. 5e). These results suggest that BMPER expression and secretion are inhibited by high glucose and palmitic acids. The inhibitory impact of AGEs on BMPER protein level, but not its mRNA level, suggests that some transcription-independent mechanisms might be also involved for the regulation of BMPER protein by hyperglycemia. To further determine whether EC is an important source for BMPER production in vivo, we generated endothelium-specific BMPER inducible knockout (eKO) mice by breeding BMPER[flox/flox] and Cdh5-CreER[+/−] mice. BMPER eKO mice, similarly to its global depletion (Fig. 1), displayed hyperinsulinemia, glucose intolerance, and insulin resistance without weight difference compared to their littermate control (eWT, BMPER[flox/flox]; Cdh5-CreER[−/−]) mice (Fig. 2). BMPER depletion in ECs also resulted in hepatic and peripheral insulin resistance, indicated by clamp and insulin signaling studies (Supplementary Fig. 6). However, no significant changes in energy expenditure, physical activity, food, and water intake were observed in BMPER eKO mice compared to eWT mice (Supplementary Fig. 7). These data suggest EC is an important source for BMPER in the periphery.

**BMPER plasma level decreases in humans with metabolic syndrome.** Since BMPER serum level was lower in BMPER iKO mice that spontaneously developed hyperinsulinemia and insulin resistance, we examined whether there is an association between BMPER level and insulin resistance. To this end, we observed that BMPER plasma level was reduced by ~50% in humans with metabolic syndrome (MS)[25], a complication often coupled with insulin resistance and increased risk for T2DM[26], compared to healthy individuals (Fig. 3a, b). In addition, BMPER levels negatively correlated with body weight and plasma levels of insulin and TGs (Fig. 3c–f), suggesting an association between decreased BMPER levels and conventional serum markers of

insulin resistance. Similarly, an ~70% decrease of BMPER plasma level was observed in high-fat diet (HFD)-fed mice (Fig. 3g, h). These results suggest that BMPER plasma level is negatively regulated by metabolic stress.

**BMPER activates insulin signaling pathway through IR.** To understand the underlying mechanism by which BMPER regulates insulin sensitivity and glucose tolerance, we focused on its impact on the canonical insulin signaling pathway that governs glucose output[27,28]. As expected, we observed BMPER increased activation of insulin signaling pathway in the liver, skeletal muscle and heart, indicated by the phosphorylation of IRS1 and AKT (Fig. 4a). To determine whether BMPER regulates insulin signaling through modulating BMP signaling pathway, we used the BMPR2 kinase-dead (BMPR2-KD) mutant construct that inhibited BMP2-induced Smad 1, 5, 8 phosphorylation in primary hepatocytes (Supplementary Fig. 8a). However, BMPR2-KD mutant failed to inhibit BMPER-promoted IRS1 phosphorylation (Fig. 4b). In addition, BMP2 or BMP4 did not increase IRS1 phosphorylation (Supplementary Fig. 8b). On the other hand, IRS1 phosphorylation induced by BMPER or insulin was blocked in IR-depleted hepatocytes isolated from IR-iKO mice (Fig. 4c). Thus, our data suggest that IR, but not BMP signaling, is required for BMPER activation of insulin signaling.

**BMPER-activated insulin signaling requires NPC1 and BMPER internalization.** To further understand how BMPER increased insulin signaling, we tested whether BMPER binds IR. However, we failed to observe the direct interaction of BMPER and recombinant IR, indicating BMPER regulates insulin pathway via unknown mediators. By performing co-immunoprecipitation-combined with liquid chromatography–mass spectrometry proteomic techniques, we identified NPC1, an endosome and lysosome-residing membrane protein[16], as an interacting protein of BMPER (Supplementary Fig. 8c, Table 1). NPC1 loss of function mutants develops devastating lysosomal cholesterol accumulation[29,30]. NPC1 also influences insulin signaling in adipocytes and NPC disease mouse models[31,32]. However, its exact role in insulin action remains unclear. Therefore, we tested the notion that NPC1 is required for BMPER to regulate insulin signaling. We confirmed NPC1 and BMPER were in the same complex and NPC1 N-terminal domain was responsible for their association (Fig. 5a, b).

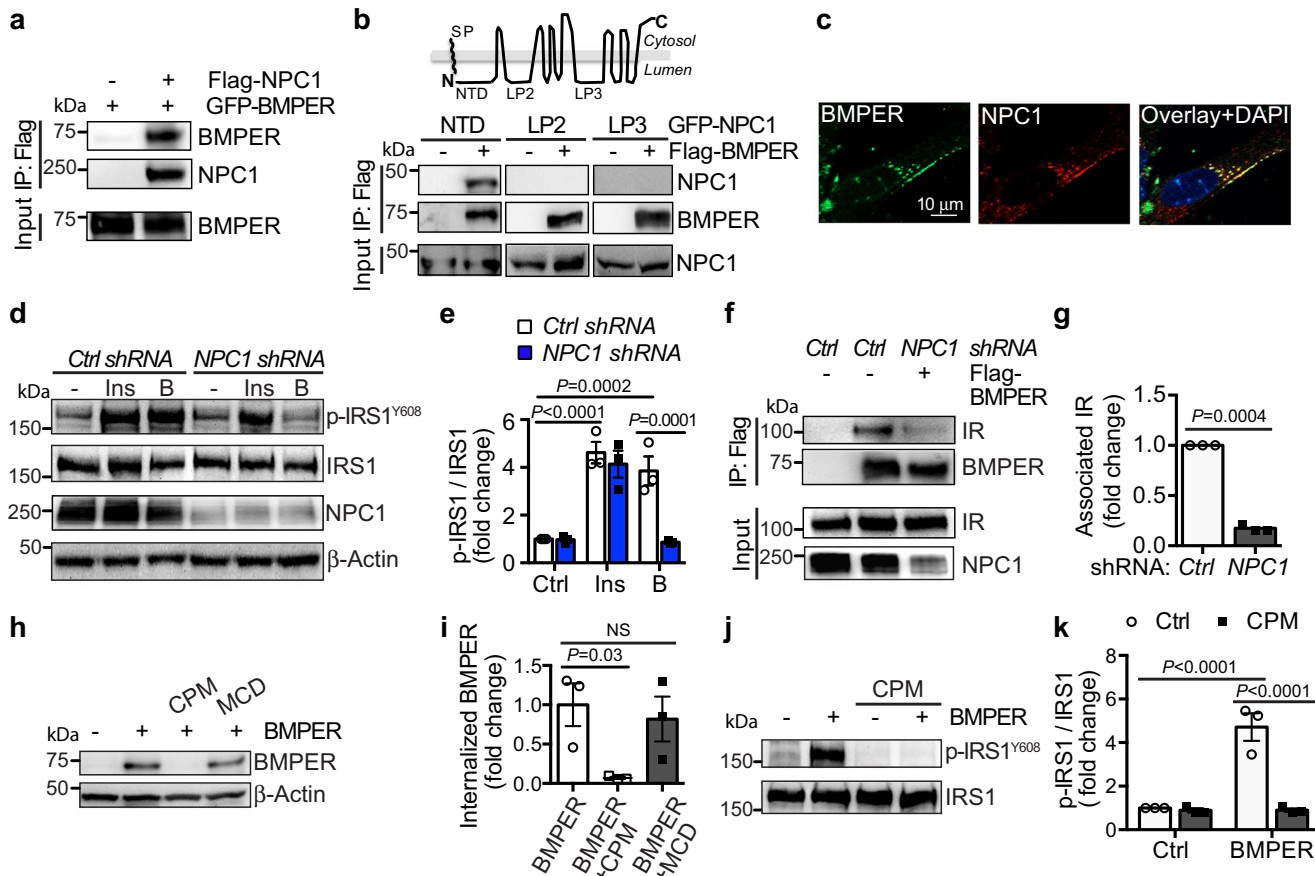

**Fig. 5 BMPER promotes insulin signaling through NPC1 and endocytosis. a** Co-immunoprecipitation (IP) of GFP-tagged BMPER and flag-tagged NPC1 in HEK293 cells. **b** Co-immunoprecipitation of GFP-tagged NPC1 constructs containing different lumenal domains (NTD, N-terminal domain, a.a. 25-164; LP2, loop 2, a.a. 371-615; LP3, loop 3, a.a. 855-1098) and flag-tagged BMPER in HEK293 cells. SP signal peptide. **c** Hepatocytes were treated with flag-tagged BMPER for 30 min and staining of BMPER and endogenous NPC1 was performed. **d, e** Hepatocytes were transduced with *NPC1* shRNA lentivirus and then treated with flag-tagged BMPER (B, 1 h) or insulin (Ins, 30 min). Western blotting was performed and band intensity was quantified (**e**). **f, g** Hepatocytes were transduced with *NPC1* shRNA lentivirus and then treated with flag-tagged BMPER (B, 1 h). Membrane fractions were separated and then subjected for IP with flag antibody. The associated IR with BMPER was quantified (**g**). **h, i** Hepatocytes were treated with chlorpromazine (CPM, 50 μM) or methyl-β-cyclodextrin (MCD, 10 mM). Thirty minutes later, cells were pulsed with flag-BMPER (100 nM) for 1 h and cell lysates were used for Western blotting. Internalized BMPER was quantified in (**i**). **j, k** Hepatocytes were treated with CPM for 30 min, and then treated with BMPER for the detection of IRS1 phosphorylation. *n* = 3 repeated experiments (**e, g, i, k**). Data are presented as mean values ± SEM. NS not significant. Analysis was two-way ANOVA (for **e, k**) or one-way ANOVA (for **i**) followed by Fisher's LSD multiple comparison test and unpaired two-tailed Student's *t* test (**g**).

In addition, BMPER interacted with endogenous NPC1 in hepatocytes and they are co-localized mainly in vesicles inside of hepatocytes (Fig. 5c, Supplementary Fig. 8d). Notably, knockdown of NPC1 markedly blocked BMPER, but not insulin-induced IRS1 phosphorylation (Fig. 5d, e). However, NPC2 knockdown did not inhibit BMPER's effect (Supplementary Fig. 8e). It suggests that NPC1 specifically mediates BMPER-promoted insulin signaling. By performing immunoprecipitation studies with membrane fractions purified from hepatocytes, we discovered that BMPER was in the same complex with endogenous IR in vivo and knockdown of NPC1 markedly blocked the BMPER and IR complex formation in hepatocytes (Fig. 5f, g). However, NPC2 knockdown or methyl-β-cyclodextrin (MCD) treatment that changes membrane cholesterol contents did not affect the interaction of BMPER and IR (Supplementary Fig. 8f, g). We also detected the association of over-expressed NPC1 and IR in HEK293 cells and that of endogenous NPC1 and IR in hepatocytes (Supplementary Fig. 8d, h). These observations suggest that NPC1 might serve as a scaffold protein for the interaction of BMPER and IR. Our previous studies demonstrate BMPER can be internalized into endothelial cells[15]. In hepatocytes, BMPER was also internalized and the internalized

BMPER was degraded with a half-life at 38.45 min during a chase period with cold media (Supplementary Fig. 8i). Its internalization was inhibited by chlorpromazine (CPM), an inhibitor of clathrin-dependent endocytosis but not by MCD, an inhibitor of the caveolin-dependent endocytosis (Fig. 5h, i). In addition, CPM abolished IRS1 phosphorylation induced by BMPER (Fig. 5j, k), suggesting BMPER internalization is required for IRS1 activation.

**BMPER supplementation improves glucose homeostasis in diabetic mice.** Since BMPER depletion resulted in defective glucose metabolism, we hypothesized that BMPER supplementation improves glucose homeostasis in diabetic mice. We injected adeno-associated virus (AAV) of BMPER, or AAV-GFP as the control, into mice and then fed them with HFD for 8 weeks. We observed AAV-BMPER injection recovered BMPER plasma level in HFD-fed mice back to the level seen in control chow-fed mice (Fig. 6a) and had no dramatic impacts on energy expenditure, physical activity, appetite, and weight gain (Supplementary Figs. 9, 10, and 11a, b). However, plasma insulin and glucose levels were decreased in AAV-BMPER-injected mice

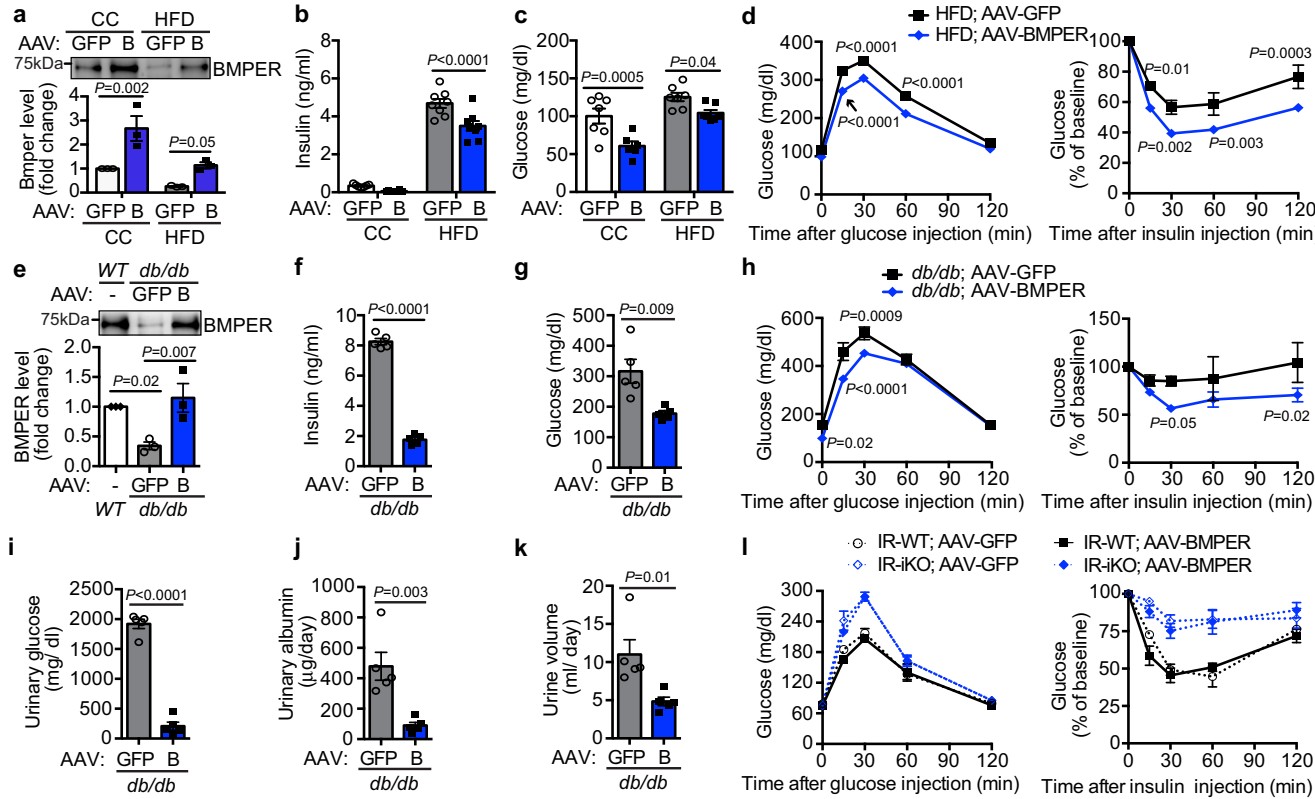

**Fig. 6 AAV-BMPER improves glucose responses in diabetic mice. a** BMPER plasma levels in C57BL/6 (WT), AAV-GFP, or AAV-BMPER (B)-injected mice that were fed HFD or CC diet. **b, c** Fasted insulin and glucose. **d** Glucose and insulin tolerance tests. **e** BMPER plasma levels in C57BL/6 (WT), AAV-GFP, or AAV-BMPER (B)-injected *db/db* mice. **f, g** Fasted insulin and glucose. **h** Glucose and insulin tolerance tests. **i** Urinary glucose levels. **j** Urinary albumin levels. **k** Urine volume. **l** Glucose and insulin tolerance tests. IR-WT, IR$^{flox/flox}$; CAG-CreER$^{-/-}$. IR-iKO, IR$^{flox/flox}$; CAG-CreER$^{+/-}$. $n = 3$ mice (**a, e**), 6–8 mice (**b–d** AAV-GFP), 6–8 mice (**b–d** AAV-BMPER), 5 mice (**f, g, i–k**), 5 mice (**h** AAV-GFP), 6–7 mice (**h** AAV-BMPER), 4 mice (**l** IR-WT;AAV-GFP, IR-iKO; AAV-GFP, IR-WT;AAV-BMPER), and 5 mice (**l** IR-iKO;AAV-BMPER). Data are presented as mean values ± SEM. Analysis was two-way ANOVA (for **a–d**, **h**, **l**) or one-way ANOVA (for **e**) followed by Fisher's LSD multiple comparison test or unpaired two-tailed Student's *t* test (**f, g, i–k**).

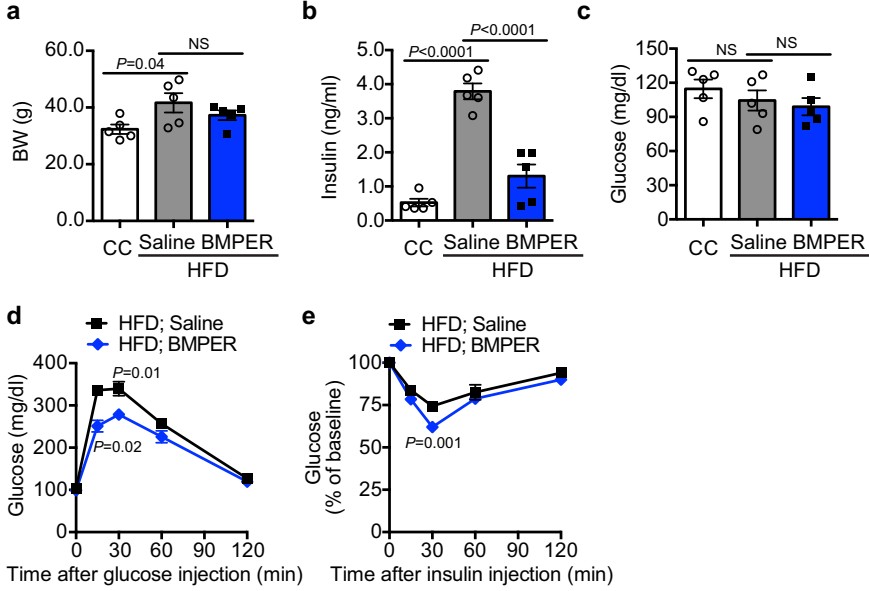

**Fig. 7 Recombinant BMPER protein improves glucose responses in DIO mice.** C57BL/6 mice were fed HFD or control chow (CC) for 4 weeks and then injected with recombinant BMPER protein at 0.1 mg/kg/mouse every other day for 6 weeks. Metabolic parameters were measured, including **a** Body weight. **b** Fasted insulin. **c** Fasted glucose. **d** Glucose tolerance tests. **e** Insulin tolerance tests. $n = 4$ mice (**a** CC), 5 mice (**a** HFD), and 5 mice (**b–e**). Data are presented as mean values ± SEM. NS not significant. Analysis was one-way ANOVA (for **a–c**) or two-way ANOVA (for **d, e**) followed by Fisher's LSD multiple comparison test.

compared to AAV-GFP group (Fig. 6b, c, Supplementary Fig. 11c, d). Moreover, responses of glucose clearance and insulin sensitivity were improved in AAV-BMPER- injected mice following HFD feeding, although no difference was observed in CC-fed mice (Fig. 6d, Supplementary Fig. 11e–g). TG and DAG contents were decreased in the liver, but not in skeletal muscle, of AAV-BMPER-injected mice following HFD feeding (Supplementary Table 2). However, no dramatic difference was observed with the response of glucose-induced insulin secretion in AAV-BMPER-injected mice compared to AAV-GFP group (Supplementary Fig. 11h).

We also tested whether BMPER alleviates insulin resistance in a Lepr[db/db] (db/db) monogenic mouse model of severe diabetes. BMPER plasma level was markedly lower than wild-type (C57BL/6) mice and AAV-BMPER injection recovered BMPER level in db/db mice (Fig. 6e). We observed no differences in food intake and body weight between AAV-BMPER and AAV-GFP-injected mice (Supplementary Fig. 12a, b). However, AAV-BMPER injection led to the normalization of hyperinsulinemia and hyperglycemia (Fig. 6f, g, Supplementary Fig. 12c, d), improved glucose clearance and insulin sensitivity, and decreased TG and DAG contents in db/db mice (Fig. 6h, Supplementary Table 2). Polyuria and glucosuria are important hallmark clinical features of T2DM. We observed dramatic decreases in urinary glucose and albumin levels and urine volume in AAV-BMPER-injected db/db mice (Fig. 6i–k).

The liver-specific IR knockout mice display severe insulin resistance and defects in insulin clearance[33]. Our signaling studies suggest that IR is required for BMPER-promoted insulin signaling (Fig. 4c). Therefore, we tested whether AAV-BMPER could improve glucose response in IR-iKO mice that developed insulin resistance and glucose intolerance (Fig. 6l). Although AAV-BMPER improved glucose response in diet-induced obese (DIO) and db/db mice (Fig. 6a–k), it failed to improve insulin resistance or glucose clearance in IR-iKO mice (Fig. 6l). It suggests that BMPER protects glucose homeostasis through an IR-dependent mechanism.

We also tested the feasibility of delivering recombinant BMPER protein in vivo. Following the injection of recombinant BMPER into HFD-fed mice, we detected significant normalization of hyperinsulinemia and improved glucose and insulin tolerance responses (Fig. 7). Taken together, our results suggest BMPER supplementation through AAV or recombinant protein delivery can improve T2DM.

## Discussion

In summary, our data unravel an unexpected role for BMPER in glucose homeostasis and indicate BMPER supplementation (i.e., gene therapy or recombinant protein delivery) can be a potential approach to treat T2DM and insulin resistance. Notably, the requirement of NPC1 and endocytosis for BMPER's action suggests a new perspective for transactivating insulin signaling. Although the exact role for NPC1 in BMPER IR complex formation and downstream insulin signaling events remains to be further characterized, our data strongly support that NPC1 can directly impact insulin signaling pathway through recruiting BMPER to IR. In addition to impact insulin signaling, our results suggest BMPER regulates gluconeogenesis, lipogenesis, and TG/DAG metabolism, which might also contribute to BMPER regulation of insulin action in the liver and other metabolic organs. Recent studies demonstrate that DAG, ceramide, and BCAA play key roles in liver and muscle insulin action[34]. Interestingly, the level of DAG, but not ceramide and BCAA, is significantly changed by BMPER in the liver and muscle (Table 1, Supplementary Table 2), suggesting BMPER specifically regulates DAG

metabolism. It was shown recently that membrane-bound DAG caused hepatic insulin resistance through PKC-mediated insulin receptor phosphorylation[35]. It raises a possible mechanism that BMPER might regulate insulin sensitivity through DAG-mediated signaling events, which warrants further investigation.

The endothelium, by virtue of its location, tightly controls metabolic exchange between the circulation and surrounding tissues. Endothelial dysfunction and inflammation have been observed during the development of T2DM and insulin resistance[6,7], however, it remains a question how important endothelial cell is in the maintenance of metabolic homeostasis and how its dysregulation contributes to metabolic dyshomeostasis. Here, our data indicate BMPER is downregulated in humans with metabolic syndrome and DIO mice (Fig. 3), suggesting endothelial secretory function might also be disrupted during metabolic stress conditions and contribute to the progression of T2DM and insulin resistance.

## Methods

**Reagents and antibodies**. All the chemicals and antibodies are listed in the main reagent table (Supplementary Table 3).

**Mice**. BMPER[flox;flox] mice were generated with standard CRISPR-cas9 gene editing technique. C57BL/6 embryos were microinjected with Cas9 protein, each guided RNA and a circular donor vector. Designed guide RNAs targeted intron 3 and intron 4 of the bmper gene for insertion of loxP sites flanking Exon 4 (Fig. 1a). Guide RNAs were evaluated in vitro using purified Cas9 and PCR-amplified target regions. Functional guide RNAs were identified and used for model generation. Two founder animals were identified by specific PCR assays as positive for both 5′ and 3′ loxP site insertions. PCR assays spanning each loxP site followed by sequencing were also performed to confirm the loxP sequences. Spanning PCR assays across both loxP sites were performed to confirm cis orientation. Further validation to rule out random integration of the donor vector was performed, including long PCR assays across 5′ and 3′ homology arms and PCR assays specific to the donor vector backbone. In addition, off-target analysis was performed on the original founders and no hits in any of the potential off-target sites screened were identified. Male chimeras were mated to wild-type C57BL/6 females to establish an isogenic line, and then backcrossed to C57BL/6 mice to obtain BMPER[flox/flox] mice on C57BL/6 background. All experiments were conducted on the resulting C57BL/6 background. Genomic DNA of BMPER[flox;flox] mice were isolated as described previously[12] and subjected for standard PCR assays to identify wild-type and targeted alleles. A PCR assay has been developed to genotype the pups. PCR samples were denatured in 95 °C for 2 min, and then subjected to 35 cycles of three-step amplification, a 30-s 94 °C denaturation, 30-s 50 °C annealing, and 1-min 72 °C extension step. A 740-bp product represents the wild-type allele and an 820-bp product (primers BMPER-SeqF and BMPER-SeqR) indicates the target allele. PCR primers: BMPER-SeqF, 5′-CGCACCCTCTAA CCTGTTCAG AC-3′; BMPER-SeqR, 5′-AGAACCACTGTTTTGCTCCAAGC-3′. The B6.Cg-Tg (CAG-cre/Esr1*) 5Amc/J (CAG-CreER[+/−]) mice were obtained from Jackson Laboratories. The Cdh5 (PAC)-CreERT2 (Cdh5-CreER[+/−]) mice were generously provided by Dr. Ralf H. Adams[36]. All mice were housed on a 12-h light/dark cycle, with food and water ad libitum. All experimental procedures on mice were performed according to the National Institutes of Health Guide for the Care and Use of Laboratory Animals and approved by the Institutional Committee for the Use of Animals in Research at Baylor College of Medicine.

We used the mating of BMPER[flox;flox] and CAG-CreER[+/−] or BMPER[flox;flox] and Cdh5-CreER[+/−] mice to generate the BMPER[flox;flox]; CAG-CreER[+/−] (WT or iKO) mice or BMPER[flox;flox]; Cdh5-CreER[+/−] (eWT or eKO) male mice for control chow (CC, 14.7% calories from fat) or high-fat diet (HFD, 60% calories from fat)-induced diabetes studies. In addition, IR-iKO and their littermate control IR-WT mice were generated from the mating of IR[flox/flox] and CAG-CreER[+/−] mice followed by tamoxifen injection. Blood serum was obtained before and after they were fed with different diets. Primary hepatocytes were isolated from 5~8 weeks mice. For insulin signaling experiments, mice were injected with insulin at 0.5 U/kg (for CC fed mice) or 1.0 U/kg (for HFD and db/db mice) and blood glucose levels were monitored. For adeno-associated virus (AAV)-transduced experiments, C57BL/6 and db/db mice at 5-weeks old were injected intravenously with the AAV-GFP or AAV-BMPER, respectively, the titer at ~10[12] per 25 g mice. C57BL/6 mice were then fed CC or HFD and db/db mice were fed CC as indicated. Metabolic studies were performed with these mice and tissues and serum were collected for further analysis.

**Subjects**. Plasma samples were obtained from the study[25] that was approved by the Institutional Review Board of Baylor College of Medicine. Male and female volunteers were recruited at the Center for Cardiometabolic Disease Prevention at

Baylor College of Medicine, Houston, Texas or by advertisement. The informed consent were obtained and all complied with the relevant ethical regulations.

**Cell lines and primary cells**. HEK293 cells were grown in DMEM supplemented with 10% FBS and antibiotics (100 U/ml penicillin, 68.6 mol/L streptomycin). Primary hepatocytes were isolated using collagenase perfusion method from C57BL/6 mice based on a previously published paper with modifications[37]. Briefly, the liver from 5~8-weeks-old mice was perfused through the inferior vena cava with 7 ml pre-warmed (37 °C) liver perfusion medium at 1 ml/min. Then, the liver was constantly digested with 5 ml pre-warmed (37 °C) collagenase (1 mg/ml) at 1 ml/min. The perfused liver was chopped gently in DMEM and centrifuged at 72 g for 2 min. Cells were washed with 20 ml hepatocyte wash medium and purified with 20 ml 45% (v/v) Percoll solution at 72 g for 2 min. Cell viability and number were measured through Trypan blue exclusion and manual counting. Hepatocytes were cultured in William's E medium containing 5% FBS, 100 units/ml of penicillin and streptomycin, and primary hepatocyte maintenance supplements at 37 °C in a 5% $CO_2$ incubator. For NPC1 or NPC2 depleted cells, primary hepatocytes were transduced with lentiviral *NPC1*, *NPC2* shRNA, or control virus for 3 days. For signaling experiment, hepatocytes were starved overnight and treated with insulin at 100 nM for 30 min and BMPER at 100 nM for 1 h in all the experiments.

**Analysis of endocrine hormones and metabolites**. For fasting blood serum, mice were fasted 4 h or overnight and then blood was collected through submandibular bleeding using a lancet. Plasma and urinary values for glucose were measured with a mouse endocrine multiplex assay, and insulin, albumin, TG, FFA, DAG, BCAA, and ceramide with commercial kits (Supplementary Table 3).

**Glucose/insulin tolerance tests**. Glucose tolerance tests (GTTs) and Insulin tolerance tests (ITTs) were performed following our established protocol[38]. Briefly, GTTs were performed after an overnight (for CC and HFD-fed mice or *db/db* mice) fasting. Blood glucose was measured before and 15, 30, 60, 120 min after an *i. p.* glucose injection (1 g/kg) with a Freestyle Glucose Monitoring System (Abbott Laboratories). ITTs were performed after 4 h fasting. Blood glucose was measured before and indicated time periods after an *i.p.* insulin injection.

**Hyperinsulinemic-euglycemic clamp studies**. The hyperinsulinemic-euglycemic clamp studies were performed in unrestrained mice using the insulin clamp technique (with constant insulin dose) in combination with [$^3$H] glucose and [$^{14}$C] 2-deoxyglucose following our established protocol. In summary, mice were cannulated as described previously[39] and allowed to recover for 4–7 days before the clamp. After an overnight fasting, mice received a primed dose of [$^3$H] glucose (10 μCi) and then a constant rate intravenous infusion (0.1 μCi/min) of [$^3$H]glucose using a syringe infusion pump for 90 min. Blood samples were collected for the determination of basal glucose production. After 90 min, mice were primed with a bolus injection of insulin followed by a 2 h continuous insulin infusion. Simultaneously, 25% glucose was infused at an adjusted rate to maintain the blood glucose level at 100–140 mg/dl. Blood glucose concentration was determined every 10 min by a glucometer. At the end of a 120-min period, blood was collected for the measurements of hepatic glucose production and peripheral glucose disposal rates. For tissue-specific uptake, we inject 2-deoxy-D-[1,-$^{14}$C] glucose (10 μCi) into bolus during hyperinsulinemic-euglycemic clamp at 45 min before the end of the clamps and collect blood sample at 5, 10, 15, 25, 35, and 45 min. At the end of the clamp, mouse tissues were harvested for the evaluation of glucose uptake. Glucose uptake in different tissues was calculated from plasma 2-[$^{14}$C] deoxyglucose profile and tissue content of [$^{14}$C] glucose-6 phosphate.

**Indirect calorimetry and metabolic cage studies**. This assay was performed following our established protocol[38] with minor modification. Mice were individually housed in metabolic chambers maintained at 20–22 °C on a 12-h light/dark cycle. Metabolic measurements (oxygen consumption, food and water intake, locomotor activity) were recorded using an Oxymax/CLAMS (Columbus Instruments) open-circuit indirect calorimetry system. Food and water intake were also monitored. To determine the amount of urinary albumin excretion, an individual mouse was separated in a metabolic cage, where urine was collected and measured for 24 h. The urinary volume and its glucose and albumin concentration were determined.

**Membrane fractionation**. Membrane fractionation was performed following our published protocol with small modification[15]. Briefly, tissue or cells were washed twice with cold PBS and suspended in buffer A (10 mM HEPES, pH 7.9; 10 mM KCl; 0.1 mM EDTA, 1 mM DTT, protease inhibitor). The mixture was homogenized and incubated on ice for 10 min, and then centrifuged at 4 °C at 10,000 g for 3 min. The supernatant was collected and further centrifuged at 100,000 g at 4 °C for 1 h. The cloudy pellet was washed and dissolved in the lysis buffer as the membrane fraction, which was used for further analysis.

**Immunoblotting and immunoprecipitation**. Immunoblotting experiments were performed based on our previously published paper[40] with slight modification. Briefly, cells were harvested in lysis buffer (1% Triton X-100, 50 mM Tris, pH 7.4,

150 mM NaCl, 10% glycerol, and protease and phosphatase inhibitors) and clarified by centrifugation at 8000 g for 5 min. Equal amounts of protein were loaded on the SDS-PAGE gel and subjected for Western blotting. For endogenous immunoprecipitation (IP) experiments, protein A/G Plus-agarose was used to pull down antibody complexes following our established methods[13]. For transient transfection, HEK293 cells were transfected with Flag-tagged NPC1, GFP-tagged BMPER plasmid, or both together with Lipofectamine 2000. Two days later, cell lysates were harvested and IPed with anti-Flag resin and precipitates were blotted with anti-GFP and anti-flag antibodies. For endogenous IP, primary hepatocytes were transduced with lentiviral control, NPC1 or NPC2 shRNA, respectively. Three days later, purified flag-tagged BMPER protein was added to the medium for 30 min. Cells were then washed with cold PBS and cross-linked with DSP at 4 °C for 2 h. Cells were harvested and membrane fractions were used for IP experiments.

**Gene expression analysis (real-time PCR)**. Total RNAs were reversely transcribed into cDNAs with iScript$^{TM}$ cDNA synthesis kit. The specific pairs of primers used for the real-time PCR are listed in Supplementary Table 4 (designed by Universal ProbeLibrary Assay Design Center tool from Roche, Indianapolis, IN). The real-time PCR was performed with FastStart Universal Probe Master mix, specific primers and probes for each gene (Universal ProbeLibrary Probes #19 for G6Pase, #49 for PEPCK, #58 for GK, #77 for SREBP1, #38 for IL1β, #6 for IL6, #25 for TNFα, #64 for β-Actin, #79 for BMPER, #20 for BMP2, #63 for BMP4, #22 for BMP6, #1 for BMP7, #38 for BMP9, #67 for BMPR2) in Roche Lightcycler 480 PCR machine. Reaction mixtures were incubated at 95 °C for 10 min followed by 55 cycles at 95 °C for 10 s and 60 °C for 30 s. β-Actin was used as the housekeeping gene.

**Statistics and reproducibility**. All the experiments were performed at least three independent experiments or sufficient sample sizes were involved. Micrographs shown in the results are representative images obtained from experiments that have been repeated independently for at least three times. No statistical methods were used to predetermine the sample size. No randomization was used as all mice used were genetically defined, inbred mice. Statistical data were drawn from normally distributed groups with similar variance between groups. Data are shown as the mean ± SEM. Differences were analyzed with Student's $t$ test, one-way or two-way ANOVA, and followed by a Fisher's LSD test unless otherwise specifically stated. Values of $P \leq 0.05$ were considered statistically significant.

**Reporting summary**. Further information on research design is available in the Nature Research Reporting Summary linked to this article.

## Data availability

The data that support the findings of this study are available within the article, its Supplementary Information files. The detailed information of key reagents is provided in Supplementary Table 3. The source data underlying Figures and Supplementary Figures are provided as a Source Data file. Source data are provided with this paper.

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

## Acknowledgements

We would like to thank the MMPC Core at BCM and NIH fund RO1DK114356 & UM1HG006348 for core services. This study was supported by grants from the US National Institutes of Health (NIH): R01HL122736 (to L.X.), R01HL112890, HL061656, and DK123186 (to X.P.). This work was also funded by an American Diabetes Association #1-18-IBS-105 (to S.M.H.).

## Author contributions

H.M., L.X., and X.P. writing the first draft, theoretical and experimental investigation, scientific discussion, revision of the manuscript. L.L., Q.F., A.A., and P.K.S. experimental investigation, scientific discussion, revision of the manuscript. H.W. and C.M.B. studies with human samples, revision of the manuscript. S.M.H. scientific discussion, revision of the manuscript.

## Competing interests

The authors declare no competing interests.
