## [Peer Review File · Nature Communications]

Reviewers' comments:

Reviewer #1 (Remarks to the Author):

In this study Mao et al. show that knockdown of BMPER in mice causes hyperinsulinemia, glucose intolerance and insulin resistance without increasing inflammation in metabolic tissues. They go on to show that BMPER can directly activate insulin signaling, which requires its internalization and interaction with Niemann-Pick C1 and that delivery of recombinant BMPER protein alleviates insulin resistance and hyperglycemia in HFD fed and db/db mice. From these results the authors conclude that BMPER may be of therapeutic potential for the treatment of type 2 diabetes.

Comments

This is an interesting study that identifies a potentially role for BMPER in the regulation of insulin action as well as reversal of insulin resistance. While the authors perform some studies in an attempt to elucidate the mechanism by which BMPER may modulate insulin action and reverse insulin resistance they do not provide a definite answer in this regard.

I have the following suggestions that would strengthen this manuscript.

1. While the authors have performed some in vitro insulin signaling studies they should also assess basal and insulin stimulated insulin signaling (IRK phosphorylation, IRS-1/IRS-2 phosphorylation, AKT phosphorylation, etc.) in the liver, muscle, heart, BAT, WAT and the endothelium in the clamp studies shown in Figure 1 as well as in the eKO/eWT mice (Supplementary Figure 5) and the AAV-BMPER /AAV-GFP mice (Supplementary Figure 7)
2. The authors should also assess metabolites (BCAAs, ceramides, diacylglycerides, etc.) that are associated with insulin resistance and which have been shown to modify insulin action in liver and skeletal muscle in the iKO and BMPER treated mice.
3. Given the increased plasma TG concentrations in the iKO mice (Supplementary Table 1) the authors should measure liver and muscle triglyceride content in the iKO and AAV-BMPER and BMPER treated mice.
4. How do the authors explain the reduced RER in the iKO mice (Supplementary Figure 2)?
5. The authors should also perform CLAMS studies in the eKO and AAV-BMPER mice.
6. How do the authors explain the link between BMPER action in the endothelium and insulin action in liver, muscle, BAT and WAT and its ability to reverse insulin resistance in these tissues?

Reviewer #2 (Remarks to the Author):

This is an interesting study showing a significant role endothelial BMPER against obesity- and diabetes-related insulin resistance in the liver. However, more mechanistic details are needed to support the role of endothelial BMPER since multiple mechanisms have been described before for hepatic insulin resistance in diabetes. Some results are rather descriptive and whether BMPER plays a generalized role in promoting insulin sensitivity in various tissues and organs (e.g., brain and skeletal muscle) in addition to the liver using glucose as the major consumption source is unknown. This reviewer has the following major comments on this work.

1. Is BMPER produced primarily by endothelial cells or other cells including hepatocytes and tissues also express this peptide?
2. What is likely causing reduced plasma level of BMPER in diabetic mice and diabetic patients? Does high glucose, fatty acids or AGEs reduce BMPER synthesis in vulture human endothelial cells or primary mouse endothelial cells?
3. Although the authors provide evidence for the involvement of NCP1, how it regulates BMPER and IR interaction is still unclear.
4. The effect of BMPER in insulin sensitivity of other metabolic organs such as skeletal muscle and adipose tissues in addition to the liver is not provided. By comparing two insulin sensitivity curves in Figure 1g and Supplemental Figure 4g, it is clear global KO of BMPER has much greater impact compared with endothelial KO of BMPER. Further study is needed to identify the other sources of

BMPER in vivo.

5. Does loss of BMPER function inhibit insulin-induced production of nitric oxide in endothelial cells via impairing insulin signaling pathway?
6. Does BMP4/2 affect insulin signaling in both endothelial cells and hepatocytes? If yes, what will be the effect of BMPER?
7. Does exogenous BMPER affect glucose-induced insulin release in pancreas?
8. Some of the data on eKO in Supplemental Figure 4 should be added to the main figures.

Reviewer #3 (Remarks to the Author):

The authors investigated the role of BMPER in insulin signalling primarily through mouse experiments. Mechanistically, the authors show that NPC1 and endocytosis may play a role in BMPER's function (Fig. 4). NPC1 was implicated in obesity in clinical studies, but the mechanism is not clear. The link between BMPER and NPC1 reported here could be interesting, but the data from this study concerning NPC1 are very weak.

1. Fig. 4a, two overexpressed proteins are used in the IP. Given the nature of NPC1, many proteins can interact with NPC1 in such an assay. There are excellent antibodies available for NPC1 so this IP needs to be repeated.
2. BMPER is a secretory protein so it should interact with the luminal domains of NPC1. There are three large domains. It would highly strengthen the paper if the authors can identify which domain interact with BMPER. NPC2 interacts with one of those domains and can serve as a good control.
3. It seems that NPC1 may serve as a scaffold facilitating BMPER-IR interaction. Does NPC1 interact with IR?
4. NPC1's impact on BMPER-IR interaction could arise from changes in membrane cholesterol. If this is the case, NPC2 should have a similar impact (fig. 4c, 4e).
5. In the text, some figures in figure 4 are labelled as figure 3.

Reviewer #1 (Remarks to the Author):

In this study Mao et al. show that knockdown of BMPER in mice causes hyperinsulinemia, glucose intolerance and insulin resistance without increasing inflammation in metabolic tissues. They go on to show that BMPER can directly activate insulin signaling, which requires its internalization and interaction with Niemann-Pick C1 and that delivery of recombinant BMPER protein alleviates insulin resistance and hyperglycemia in HFD fed and db/db mice. From these results the authors conclude that BMPER may be of therapeutic potential for the treatment of type 2 diabetes.

Comments

This is an interesting study that identifies a potentially role for BMPER in the regulation of insulin action as well as reversal of insulin resistance. While the authors perform some studies in an attempt to elucidate the mechanism by which BMPER may modulate insulin action and reverse insulin resistance they do not provide a definite answer in this regard.

I have the following suggestions that would strengthen this manuscript.

1. While the authors have performed some in vitro insulin signaling studies, they should also assess basal and insulin stimulated insulin signaling (IRK phosphorylation, IRS-1/IRS-2 phosphorylation, AKT phosphorylation, etc.) in the liver, muscle, heart, BAT, WAT and the endothelium in the clamp studies shown in Figure 1 as well as in the eKO/eWT mice (Supplementary Figure 5) and the AAV-BMPER /AAV-GFP mice (Supplementary Figure 7).

We really appreciate the great idea that the Reviewer provided to us. To evaluate insulin-stimulated insulin signaling in metabolic tissues, we injected BMPER WT and iKO mice with insulin and collected these metabolic tissues (liver, gastrocnemius muscle, heart, BAT and WAT) for Western blotting assays. As expected, insulin increased the phosphorylation of IRS1 and AKT (Fig. 11, Supplementary Fig. 1), indicating the activation of insulin signaling by injected insulin. However, the activity of insulin pathway was decreased in BMPER iKO tissues. We performed similar experiments with BMPER eWT and eKO mice and observed decreases of insulin signaling activity in the metabolic tissues of BMPER eKO mice, compared to BMPER eWT mice (Supplementary Fig. 6e). On the other hand, the activity of insulin pathway was increased in metabolic tissues of AAV-BMPER-injected mice, compared to AAV-GFP group (Supplementary Fig. 11e). These results strongly suggest that BMPER regulates insulin sensitivity through impacting insulin signaling.

To study the insulin signaling in endothelial cells, we need isolate ECs in insulin-injected mice. However, current techniques for EC isolation and purification from mice take a whole day. During this long procedure, the phosphorylatory status of insulin signaling mediators will change dramatically. Instead of ECs, we evaluated insulin responses with aortas, which contains ECs but also other cells such as smooth muscle cells and fibroblasts. We noticed that the activity of insulin pathway was decreased in aortas of BMPER iKO and eKO mice, compared to their control aortas (Rebuttal Fig. 1a-b). Conversely, AAV-BMPER-injected mice displayed increased insulin responses compared to AAV-GFP group (Rebuttal Fig. 1c). We also treated mouse primary ECs with insulin and discovered that insulin-stimulated IRS1 phosphorylation was inhibited in BMPER knockdown ECs (Rebuttal Fig. 1d). Taken together, insulin-stimulated insulin signaling was regulated by BMPER in metabolic tissues and ECs.

2. The authors should also assess metabolites (BCAAs, ceramides, diacylglycerides, etc.) that are associated with insulin resistance and which have been shown to modify insulin action in liver and skeletal muscle in the iKO and BMPER treated mice.

We appreciate this great suggestion. To determine the changes of metabolites, we homogenized liver and gastrocnemius muscle (GM) lysates of WT and BMPER iKO mice and performed biochemical assays for diacylglyceride (DAG), branched chain amino acid (BCAA) and ceramide. BMPER iKO mice displayed significant increases in liver DAG contents (Supplementary Table 1). On the other hand, DAG contents were significant lower in liver and GM of AAV-BMPER-injected *db/db* mice compared to AAV-GFP group (Supplementary Table 3). DAG contents were also decreased in liver of AAV-BMPER-injected mice following HFD feeding compared to AAV-GFP group (Supplementary Table 3). However, we did not observe significant changes in BCAA and ceramide contents in BMPER iKO mice compared to WT

mice, and in AAV-BMPER-injected mice compared to AAV-GFP group (Supplementary Table 1, 3). These data suggest that BMPER specifically regulates DAG metabolism in liver and skeletal muscle tissues.

3. Given the increased plasma TG concentrations in the iKO mice (Supplementary Table 1) the authors should measure liver and muscle triglyceride content in the iKO and AAV-BMPER and BMPER treated mice.

This is a wonderful suggestion! We homogenized liver and GM lysates of WT and BMPER iKO mice and measured triglyceride (TG) levels. BMPER iKO mice displayed a significant increase in liver TG content and a non-significant trend of elevated GM TG content (Supplementary Table 1). On the other hand, AAV-BMPER-injected mice displayed decreased TG contents in the liver, but not in GM (Supplementary Table 3). These results suggest that BMPER regulates TG metabolism in the liver.

4. How do the authors explain the reduced RER in the iKO mice (Supplementary Figure 2)?

A great question. As the Reviewer noticed, we were also puzzled by this mild reduction in RER (Supplementary Fig. 3c), the ratio of CO₂ production to O₂ consumption [VCO₂:VO₂]. Since RER reflects the ratio of carbohydrate to fatty acid oxidation, this observation suggests that BMPER iKO mice use a relative greater ratio of fatty acids as a fuel source than WT mice. However, the underlying cause of this RER reduction is still unclear. Since increased physical activity has been reported to decrease RER¹ and BMPER iKO mice displayed increased physical activity (Supplementary Fig. 3e-f), we speculate that the reduced RER in BMPER iKO mice could be due to their increased physical activity. In addition, the reduced RER could be affected by other metabolic changes, including fatty acid metabolism, glycogen storage, etc. These unanswered questions still need further investigation.

5. The authors should also perform CLAMS studies in the eKO and AAV-BMPER mice.

Thank you very much for this thoughtful comment. As the Reviewer suggested, we performed CLAMS assays with BMPER eKO and AAV-BMPER-injected mice. We observed that BMPER depletion in ECs or overexpression of BMPER did not dramatically change energy expenditure, physical activity, food and water intake except decreases of heat generation in AAV-BMPER-inject mice following HFD feeding (Supplemental Figure 7, 9, 10).

6. How do the authors explain the link between BMPER action in the endothelium and insulin action in liver, muscle, BAT and WAT and its ability to reverse insulin resistance in these tissues?

This is a great question. Reports show that endothelial insulin signaling pathway contribute to the transport of insulin across the endothelial barrier^{2,3}. We observed that BMPER knockdown inhibited the activity of insulin-stimulated IRS1 phosphorylation and BMPER treatment increased IRS1 phosphorylation in ECs (Rebuttal Fig. 2a-b). It raises a hypothesis that BMPER might regulate trans-endothelial transport of insulin, which might indirectly contribute to the regulation of insulin sensitivity in

metabolic tissues (i.e. liver, muscle, BAT and WAT). In addition, our previous reports suggest BMPER regulates vascular tone, endothelial inflammation and angiogenesis⁴⁻⁶, which might also indirectly contribute to the regulation of insulin action in metabolic tissues. However, the detailed impacts of BMPER-dependent endothelial events on insulin resistance still need further studies in the future.

Reviewer #2 (Remarks to the Author): *This is an interesting study showing a significant role endothelial BMPER against obesity- and diabetes-related insulin resistance in the liver. However, more mechanistic details are needed to support the role of endothelial BMPER since multiple mechanisms have been described before for hepatic insulin resistance in diabetes. Some results are rather descriptive and whether BMPER plays a generalized role in promoting insulin sensitivity in various tissues and organs (e.g., brain and skeletal muscle) in addition to the liver using glucose as the major consumption source is unknown. This reviewer has the following major comments on this work.*

We appreciate the Reviewer's great comments. We performed more mechanistic studies to understand how insulin-stimulated insulin signaling was impacted by BMPER depletion or overexpression. In summary, multiple tissues and organs (i.e. liver, skeletal muscle, BAT, heart, etc.) in BMPER iKO, eKO and AAV-BMPER-injected mice displayed altered activity of insulin-induced insulin signaling pathway (Fig. 11, Supplementary Fig. 6e, 11e). Our data further suggest that BMPER might play a generalized role in promoting insulin sensitivity in multiple tissues and organs.

1. Is BMPER produced primarily by endothelial cells or other cells including hepatocytes and tissues also express this peptide?

We appreciate this great suggestion. To compare the expression of BMPER in endothelial cells, hepatocytes and a variety of tissues, we isolated RNAs from different cells and tissues and evaluated the mRNA level of BMPER. As shown in Supplemental Fig. 5a, BMPER expression was highly enriched in metabolic tissues such as BAT, GM, liver and WAT besides of bone and lung. The enriched expression of BMPER in metabolic tissues suggests BMPER might play a key role in metabolic homeostasis. In addition, BMPER expression was >20-fold higher in a variety of primary ECs than hepatocytes and mouse embryonic fibroblasts (MEFs, Supplementary Fig. 5b). It further suggests that EC is an important source for BMPER secretion.

2. What is likely causing reduced plasma level of BMPER in diabetic mice and diabetic patients? Does high glucose, fatty acids or AGEs reduce BMPER synthesis in vulture human endothelial cells or primary mouse endothelial cells?

This is a very interesting question! As the Reviewer suggested, we investigated how metabolic stress affects BMPER expression in ECs. As shown in Supplemental Fig. 5c, treatments of high glucose, AGEs and palmitic acids reduced BMPER mRNA levels in primary mouse ECs, which might explain why the plasma level of BMPER in diabetic mice was decreased. However, the underlying transcriptional mechanisms for BMPER down-regulation by diabetes in ECs still need further investigation.

3. Although the authors provide evidence for the involvement of NPC1, how it regulates BMPER and IR interaction is still unclear.

This is really a great question. To further understand the role of NPC1, we performed biochemical studies to evaluate the interaction of NPC1 and BMPER. First, we characterized the domain of NPC1 that is responsible for BMPER interaction. We generated deletion mutation constructs containing three luminal domains of NPC1 and performed co-IP experiments with BMPER. Our results (Fig. 5b) demonstrated that

the N-terminal domain of NPC1, but not Loop 2 or Loop 3, is required for the interaction with BMPER. Next, we performed IP experiments with NPC1 antibody to enrich endogenous NPC1-interacting proteins in hepatocytes. As shown in Supplemental Fig. 8d, we detected BMPER in the complex with endogenous NPC1. We also determined the effects of NPC2 knockdown and methyl- β -cyclodextrin (MCD) treatment that change membrane cholesterol contents on BMPER-IR interaction. Interestingly, neither NPC2 knockdown nor MCD impacted the interaction of BMPER and IR (Supplementary Fig. 7f-g). These results suggest that NPC1 specifically regulates the interaction of BMPER and IR. By performing immunoprecipitation assays, we detected the association of overexpressed NPC1 and IR in HEK293 cells and that of endogenous NPC1 and IR in hepatocytes (Supplementary Fig. 8d, h). These observations suggest that NPC1 might serve as a scaffold protein for the interaction of BMPER and IR.

4. *The effect of BMPER in insulin sensitivity of other metabolic organs such as skeletal muscle and adipose tissues in addition to the liver is not provided. By comparing two insulin sensitivity curves in Figure 1g and Supplemental Figure 4g, it is clear global KO of BMPER has much greater impact compared with endothelial KO of BMPER. Further study is needed to identify the other sources of BMPER in vivo.*

We totally agree with the Reviewer that global KO of BMPER has much greater impact compared with endothelial KO of BMPER. As mentioned in the answer to Question #1 of this Reviewer, we investigated the expression pattern of BMPER and discovered its expression in ECs was more than 20-fold higher than hepatocytes and MEFs. More interestingly, BMPER was abundant in metabolic tissues-BAT, GM, liver and WAT and others such as bone and lung, which suggests that BMPER source is not limited to ECs. The identification of other sources of BMPER will become one of our future research directions. Nevertheless, our data with BMPER eKO mice suggest that EC is an important source for BMPER *in vivo*.

5. *Does loss of BMPER function inhibit insulin-induced production of nitric oxide in endothelial cells via impairing insulin signaling pathway?*

This is a great question. We studied insulin-induced eNOS activation in BMPER-depleted ECs. As shown in Rebuttal Fig. 3a, insulin treatment increased activity of eNOS and IRS1 in ctrl siRNA-transfected ECs. However, BMPER knockdown inhibited their activation. Next, we depleted IR in ECs and observed that BMPER-increased eNOS and IRS1 activity was inhibited in IR-depleted ECs (Rebuttal Fig. 3b). Therefore, we hypothesize that BMPER depletion inhibits insulin-induced production of nitric oxide in ECs likely through impairing insulin signaling pathway.

6. *Does BMP4/2 affect insulin signaling in both endothelial cells and hepatocytes? If yes, what will be the effect of BMPER?*

To answer this interesting question, we treated cultured primary ECs and hepatocytes with BMP4/2 and evaluated the activity of insulin signaling pathway. As expected, both BMPER and insulin increased IRS1 phosphorylation, indicating the activation of insulin signaling pathway (Supplementary Fig. 8b, Rebuttal

Fig. 4). However, either BMP2 or BMP4 failed to increase IRS1 phosphorylation in ECs and hepatocytes, suggesting BMP4/2 has no promoting effects on insulin signaling.

7. Does exogenous BMPER affect glucose-induced insulin release in pancreas?

We appreciate the reviewer's great question. We performed glucose tolerance tests and measured insulin levels following glucose challenge. In AAV-BMPER-injected mice, insulin level was quickly increased at a similar rate as that in AAV-GFP group (Supplementary Fig. 11h). In HFD-fed mice, glucose still induced insulin secretion in similar fashion in AAV-BMPER and GFP groups, even though their basal levels of insulin were different. These results suggest that BMPER might not affect glucose-induced insulin release in pancreas.

8. Some of the data on eKO in Supplemental Figure 4 should be added to the main figures.

This suggestion is great. We have included this figure as Figure 2 in the main figure section.

Reviewer #3 (Remarks to the Author): *The authors investigated the role of BMPER in insulin signalling primarily through mouse experiments. Mechanistically, the authors show that NPC1 and endocytosis may play a role in BMPER's function (Fig. 4). NPC1 was implicated in obesity in clinical studies, but the mechanism is not clear. The link between BMPER and NPC1 reported here could be interesting, but the data from this study concerning NPC1 are very weak.*

We thank the Reviewer for the thoughtful comments! We performed further mechanistic studies to understand how NPC1 regulates BMPER-induced insulin signaling in hepatocytes. First, we characterized the binding domain of NPC1 and identified its N-terminal domain was responsible for BMPER interaction (Fig. 5b). Next, we observed that BMPER and endogenous IR and NPC1 formed a multi-protein complex in hepatocytes (Supplementary Fig. 8d). In addition, NPC2 knockdown or methyl- β -cyclodextrin (MCD) treatment did not interfere with the complex formation of BMPER and IR (Supplementary Fig. 8f, g). Compared to NPC1 knockdown, NPC2 knockdown failed to block the activation of insulin signaling upon BMPER treatment (Supplementary Fig. 8e). Taken together, we hypothesize that NPC1 might serve as a scaffold protein for the complex formation between IR and BMPER and cholesterol might not significantly impact their interaction.

1. Fig. 4a, two overexpressed proteins are used in the IP. Given the nature of NPC1, many proteins can interact with NPC1 in such an assay. There are excellent antibodies available for NPC1 so this IP needs to be repeated.

Thank you very much for your great suggestion. We performed IP experiments with NPC1 antibody to enrich endogenous NPC1-interacting proteins in hepatocytes. As shown in Supplemental Fig. 8d, we detected BMPER and endogenous IR were in the same complex with endogenous NPC1. It further suggests that BMPER, IR and NPC1 can form a multi-protein complex in hepatocytes.

2. BMPER is a secretory protein so it should interact with the luminal domains of NPC1. There are three

large domains. It would highly strengthen the paper if the authors can identify which domain interacts with BMPER. NPC2 interacts with one of those domains and can serve as a good control.

This is a great point. To characterize the domain of NPC1 that is responsible for BMPER interaction, we generated deletion mutation constructs containing three luminal domains of NPC1 and performed co-IP experiments with BMPER. Our results (Fig. 5b) demonstrated that the N-terminal domain of NPC1, but not Loop 2 or Loop 3, is required for the interaction with BMPER.

3. It seems that NPC1 may serve as a scaffold facilitating BMPER-IR interaction. Does NPC1 interact with IR?

To answer this question, we first tested the binding of overexpressed NPC1 and IR in HEK293 cells. By performing co-IP experiments, we observed their interaction (Supplementary Fig. 8h). Next, we performed IP experiments with NPC1 antibody to determine whether IR and BMPER can bind to endogenous NPC1. As shown in Supplementary Fig. 8d, both BMPER and IR were detected in the NPC1-enriched protein fraction of hepatocytes. These results suggest that NPC1 might serve as a scaffold protein for the formation of BMPER and IR complex.

4. NPC1's impact on BMPER-IR interaction could arise from changes in membrane cholesterol. If this is the case, NPC2 should have a similar impact (fig. 4c, 4e).

Wonderful comments! We determined the effect of NPC2 on the BMPER-IR complex formation. Hepatocytes were transduced with NPC2 shRNA lentiviral particles and then evaluated the complex formation of BMPER and IR. Interestingly, NPC2 knockdown did not interfere with the protein complex formed between IR and BMPER (Supplementary Fig. 8f). To further understand the role of membrane cholesterol, we also pretreated hepatocytes with MCD and evaluated the BMPER-IR complex formation. Similar as NPC2 knockdown, MCD did not impact their complex formation (Supplementary Fig. 8g). These results suggest that cholesterol does not significantly impact the complex formation between BMPER and IR.

5. In the text, some figures in figure 4 are labelled as figure 3.

We would like to apologize for our oversight and these figure labels have been corrected in our revised manuscript.

References

1. O'Neal, T.J., Friend, D.M., Guo, J., Hall, K.D. & Kravitz, A.V. Increases in physical activity result in diminishing increments in daily energy expenditure in mice. *Curr Biol* **27**, 423-430 (2017).
2. Konishi, M., *et al.* Endothelial insulin receptors differentially control insulin signaling kinetics in peripheral tissues and brain of mice. *Proc Natl Acad Sci U S A* **114**, E8478-E8487 (2017).
3. Wang, H., Wang, A.X., Liu, Z. & Barrett, E.J. Insulin signaling stimulates insulin transport by bovine aortic endothelial cells. *Diabetes* **57**, 540-547 (2008).
4. Lockyer, P., *et al.* LRP1-dependent BMPER signaling regulates lipopolysaccharide-induced vascular inflammation. *Arterioscler Thromb Vasc Biol* **37**, 1524-1535 (2017).
5. Pi, X., *et al.* Bmper inhibits endothelial expression of inflammatory adhesion molecules and protects against atherosclerosis. *Arterioscler Thromb Vasc Biol* **32**, 2214-2222 (2012).
6. Pi, X., *et al.* LRP1-dependent endocytic mechanism governs the signaling output of the bmp system in endothelial cells and in angiogenesis. *Circ Res* **111**, 564-574 (2012).

REVIEWER COMMENTS

Reviewer #1 (Remarks to the Author):

The authors have performed additional studies and have done a nice job addressing my previous comments. They now show that BMPER specifically regulates DAG metabolism in liver and skeletal muscle, which in turn tracks with insulin signaling and insulin action. In contrast they find no effects of BMPER on tissue ceramide content or plasma BCAA concentrations. These DAG data are important new data and belong in the main figures. Furthermore these new DAG results need to be discussed in the context of previous studies that have demonstrated a key role for DAGs in liver and muscle in regulating insulin signaling and insulin action in liver and skeletal muscle (Petersen et al., *Physiological Reviews* 2018, Lyu et al. *Cell Metabolism* 2020). It would also be important to discuss the lack of relationship between tissue ceramide content and BCAAs with insulin resistance in the different BMPER models, given their putative roles in causing insulin resistance.

Reviewer #2 (Remarks to the Author):

Overall, the authors were very responsive to all my eight questions with additional results from new experiments.

Some remaining minor points that still require authors' attention.

1. The authors are advised to provide the BMPER protein levels in ECs following treatment with high glucose, AGEs and palmitic acid, together with mRNA measurement for Supplemental Fig. 5c to validate these metabolic stress factors can indeed reduce BMPER production and possible secretion as well if the BMPER level in culture medium is also determined.
2. Although the BMPER mRNA level is 20 times higher in ECs than hepatocytes, the authors should also provide the protein levels in these two tissues and the fold difference may not be the same compared to mRNA difference.

Reviewer #3 (Remarks to the Author):

The authors have addressed my concerns with a number of new experiments. Data in figure 5b look convincing.

There are however some suggestions to further improve the paper:

1. Molecular weight needs to be indicated in all western blots
2. The NPC1 field has progressed tremendously in the last few years. Some of the references in the manuscript on NPC1 are outdated and inappropriate. Ref. 16 is not necessary. A recent review on NPC1 should be added PMID:30710017. Also, a major recent progress elucidating NPC1 and NPC2 structure and interaction should be cited PMID: 32544384.

Reviewer #1 (Remarks to the Author):

The authors have performed additional studies and have done a nice job addressing my previous comments. They now show that BMPER specifically regulates DAG metabolism in liver and skeletal muscle, which in turn tracks with insulin signaling and insulin action. In contrast they find no effects of BMPER on tissue ceramide content or plasma BCAA concentrations. These DAG data are important new data and belong in the main figures. Furthermore these new DAG results need to be discussed in the context of previous studies that have demonstrated a key role for DAGs in liver and muscle in regulating insulin signaling and insulin action in liver and skeletal muscle (Petersen et al., Physiological Reviews 2018, Lyu et al. Cell Metabolism 2020). It would also be important to discuss the lack of relationship between tissue ceramide content and BCAAs with insulin resistance in the different BMPER models, given their putative roles in causing insulin resistance.

We appreciate the great suggestions from the Reviewer. These DAG data are important new data and have been included in the main Table (revised Table 1).

In addition, the role of DAG in BMPER-regulated insulin action in liver and skeletal muscle and the lack of relationship between ceramide and BCAA contents with IR in BMPER models are further discussed in the Discussion section (Page 11).

“In addition to impact insulin signaling, our results suggest BMPER regulates gluconeogenesis, lipogenesis and TG/DAG metabolism, which might also contribute to BMPER regulation of insulin action in liver and other metabolic organs. Recent studies demonstrate that DAG, ceramide and BCAA play key roles in liver and muscle insulin action³⁴. Interestingly, the level of DAG, but not ceramide and BCAA, is significantly changed by BMPER in liver and muscle (Table 1, Supplementary Table 2), suggesting BMPER specifically regulates DAG metabolism. It was shown recently that membrane-bound DAG caused hepatic insulin resistance through PKC-mediated insulin receptor phosphorylation³⁵. It raises a possible mechanism that BMPER might regulate insulin sensitivity through DAG-mediated signaling events, which warrants further investigation.”

Reviewer #2 (Remarks to the Author):

Overall, the authors were very responsive to all my eight questions with additional results from new experiments. Some remaining minor points that still require authors' attention.

1. The authors are advised to provide the BMPER protein levels in ECs following treatment with high glucose, AGEs and palmitic acid, together with mRNA measurement for Supplemental Fig. 5c to validate these metabolic stress factors can indeed reduce BMPER production and possible secretion as well if the BMPER level in culture medium is also determined.

This is a great comment. As the Reviewer suggested, we measured BMPER mRNA level in EC lysates and its protein level in cultured medium following treatments of high glucose, AGE and palmitic acid. As expected, we detected a reduction of BMPER protein levels in EC lysates and conditioned media following treatments of palmitic acids, high glucose or advanced glycation end products (AGEs)-treated ECs (Supplementary Fig. 5d). In addition, BMPER mRNA levels were decreased by treatments of palmitic acid and high glucose, but not AGEs (Supplementary Fig. 5e). These results suggest that BMPER expression and secretion are inhibited by high glucose and palmitic acids. The inhibitory impact of AGEs on BMPER protein level, but not its mRNA level, suggests some transcription-independent mechanisms might be also involved for the regulation of BMPER protein by hyperglycemia.

2. Although the BMPER mRNA level is 20 times higher in ECs than hepatocytes, the authors should also provide the protein levels in these two tissues and the fold difference may not be the same compared to mRNA difference.

As the Reviewer suggested, we evaluated BMPER protein levels in ECs and hepatocytes. Compared to ECs, BMPER level is 4-fold lower in hepatocytes (Supplementary Fig. 5c). Given BMPER could be internalized into both ECs and hepatocytes, BMPER protein contents are a comprehensive output of its expression, degradation, internalization and secretion. This smaller difference of BMPER protein levels between ECs and hepatocytes

than that at mRNA level (Supplementary Fig. 5b) could be due to additional impacts from BMPER degradation, internalization and/or secretion.

Reviewer #3 (Remarks to the Author):

The authors have addressed my concerns with a number of new experiments. Data in figure 5b look convincing. There are however some suggestions to further improve the paper:

1. Molecular weight needs to be indicated in all western blots

As the reviewer suggested, molecular weight is indicated in all the Western blots now.

2. The NPC1 field has progressed tremendously in the last few years. Some of the references in the manuscript on NPC1 are outdated and inappropriate. Ref. 16 is not necessary. A recent review on NPC1 should be added PMID:30710017. Also, a major recent progress elucidating NPC1 and NPC2 structure and interaction should be cited PMID: 32544384.

Thank you very much for the suggestions! Ref. 16 is removed and the mentioned two references are included in the revised manuscript.

REVIEWERS' COMMENTS

Reviewer #2 (Remarks to the Author):

The authors addressed my remaining points satisfactorily with addition of some new results in the supplemental figures. I have no further comments.

REVIEWERS' COMMENTS

Reviewer #2 (Remarks to the Author):

The authors addressed my remaining points satisfactorily with addition of some new results in the supplemental figures. I have no further comments.

We appreciate these reviewer comments.